# How does the selection of national development zones affect urban green innovation?-Evidence from China

Xiaolin Yu[1]*, Kai Wan[2]

1 School of Business and Management, Zhongnan University of Economics and Law, Wuhan, China,
2 School of Economics, Shanghai University, Shanghai, China

* yuxiaolindmw@163.com

## Abstract

The launch of the selection process for National Development Zones(NDZs) marked a fundamental change in the construction of development zones, making it an essential position for local authorities to implement high-quality development. Based on the data of prefecture-level cities in China from 2000 to 2018, this paper examines the impact and mechanism of selecting NDZs on urban green innovation through a double-difference spatial durbin model using the selection of NDZs as a "quasi-natural experiment". The study finds that the selection of NDZs can promote green innovation in cities and has a significant window-radiating effect. The heterogeneity test results show that the implementation of the selection policy for development zones in non-old industrial cities, large and medium-sized cities, cities with easy access to transportation, and cities with high market orientation are more likely to promote urban green innovation. At the same time, the higher the level of government governance and the better the level of economic development of the development zones, the more it helps to realize the effects of the selection policy. The results of the mechanism test show that the selection of NDZs has a positive impact on urban green innovation through environmental regulation effects, resource allocation effects, and policy amplification effects.

## 1. Introduction

Since the establishment of China's first Economic and Technological Development Zone (ETDZ) in 1984, the zone has made a remarkable contribution to the country's economic growth. However, as China's economy enters a new normal phase, the 14th Five-Year Plan, no longer sets specific growth targets for the first time, but takes "Innovation Driven, Green Development" as a vital starting point for high-quality development. The 2019 Guidance on Building a Market-Oriented Green Technology Innovation System emphasizes that green technology innovation is increasingly becoming an important driving force for green development, and is an crucial support for fighting the tough battle of pollution prevention, advancing ecological civilization and promoting high-quality development. As the center of China's

**Data Availability Statement:** Data files are available from the Dryad database (https://datadryad.org/stash/dataset/doi:10.5061/dryad.8gtht76rp).

**Funding:** Our research was supported by the Chinese Ministry of Education, project number:

20YJA790055. The funders had no role in study design, data collection and analysis, decision to publish, or preparation of the manuscript.

**Competing interests:** We declare that we have no financial and personal relationships with other people or organizations that can inappropriately influence our work, there is no professional or other personal interest of any nature or kind in any product, service and company.

reaction against climate change, only by implementing the concept of green development and continuously improving the level of green innovation can China's economy shift from a crude factor-driven approach to an intensive all-factor-driven approach, and ultimately achieve high-quality economic development. However, studies have shown that the level of green innovation in Chinese cities remains relatively low [1]. Then, as the pioneer of implementing the regional development strategy, how can the development zone continue to play the role of "leader" in this critical period? This is undoubtedly a significant issue under the backdrop of tight resource constraints and sluggish economic growth.

A large number of early studies evaluated the policy effects of China's development zones with different economic functions, including economic and technological development zones and high-tech industrial development zones. By comparing regions with or without development zones, it was found that development zones were more advantageous as a place-based industrial policy in terms of economic development and attracting foreign investment [2,3]. With the continuous improvement of data and methods, the subsequent studies mostly implied the establishment of development zones as a policy shock, and adopted the method of Difference-in-Difference (DID) to explored the impact of development zone policies. Some of these studies, based on the enterprise level, have argued that the establishment of development zones will produce selection and agglomeration effects, which will not only significantly expand the production scale of enterprises within the zone, but also increase the total factor productivity of surrounding enterprises, however, this spillover effect decreases with the increase in distance of the zone and increases with the increase in density of the zone [4–6]. Another part of the study, based on the regional level, found that the establishment of development zones achieved agglomeration economies and helped to increase the size of cities FDI and exports, thus this policy effect expanded over time [7].Alder confirmed that the establishment of development zones not only significantly enhances the economic development of cities, but also promotes total factor productivity, human capital investment and capital accumulation, with significant spillover effects. However, in contrast, Zheng argueed that although development zone policies have a positive impact on areas with high levels of economic development, this effect shows a marginal diminishing effect and does not contribute to economic growth overall. And this view also supported by Mendoza [8,9]. Also, Givord pointed out that the creation of establishment of zones can have a disincentive effect on surrounding enterprises [10].

As can be seen, while most studies affirm the positive impact of development zone policies, they do not reach a consistent conclusion, and this paper argues that the reason for this disagreement is that existing studies ignores the role of policy selection [11]. Alder pointed out that the impact of development zones at different levels on the regional economy varies, with the establishment of NDZs promote economic growth significantly, while provincial-level development zones do not dramatically contribute to it [8]. weighting on the process of development zones in practice, it can be seen that the most obvious feature of the policy of upgrading provincial development zones, which was launched in 2009, is that the areas were first approved by the provincial government to become provincial development zones and then submitted an application to the central government, which in turn was selected as a NDZ. Compared to provincial-level development zones, NDZs are larger in scale, serve the development strategy at national level, and have obvious advantages in terms of tax incentives and government subsidies. This shows that the establishment and the selection of a development zone are two completely different processes. The establishment of a development zone is a process from scratch, while the selection of a NDZ is a process of selection among the already established provincial development zones. In China's upward responsibility system, then, the policy of selecting NDZs will not only make local governments strive for the "National Permit"

beforehand, but will also strengthen their policies afterwards to avoid being "taken off the list". Therefore, perhaps the most important difference between national and provincial development zones lies in the strengths of policy support. In addition, the existing literature has focused on the economic benefits of development zone policies, but an important challenge for China in achieving quality development is how to achieve economic benefits while improving environmental benefits. Meanwhile, green innovation activities, as a combination of environmental protection and economic development, are considered an important strategy for achieving sustainable development [12–14]. Previous studies have suggested ways to improve green innovation in terms of environmental regulation, government systems, foreign investment and the level of economic development, with government systems considered to be an important factor driving green innovation in cities [15–19]. As an important institutional arrangement in China, NDZ selection, there is little literature discussing its impact on urban green innovation and its mechanism. Therefore, this study focuses on the changes in the level of urban green innovation in provincial development zones after they have been selected as NDZs. On the one hand, it ensures the randomness of the selection of development zones in the sample, thus mitigating the bias of sample selection; on the other hand, it can provide useful reference for improving the mechanism of green innovation policy formulation and practice.

The significant contribution of this paper is threefold: (1) Unlike the existing literature, which focuses on the economic benefits to cities or enterprises after the establishment of a development zone, this paper argues that the establishment and upgrading of development zones are fundamentally different. Moreover, the selection policy for NDZs is more random at the city level, making it more reasonable to choose a "selection" policy than establishment from the perspective of sample selection. To this end, this paper focuses on the impact of NDZs selection on urban green innovation before and after the selection, coordinates the innovation effect and environmental effect of NDZ selection, and provides an explanation for the effectiveness of the selection policy of NDZ. (2) Unlike most studies that focus on the local impact of development zones, this paper argues that the selection of NDZs provides sufficient incentive for local governments to compete, resulting in not only a local impact but also a spill-over effect. (3) The literature has mainly explained the influence mechanism of development zones from the perspective of "agglomeration effect" and "selection effect", ignoring the competitive behavior of local governments triggered by merit-based industrial policies [13,20]. Therefore, this paper explains the mechanism of selecting NDZs from three perspectives:environmental regulation effect, resource allocation effect, and policy amplification effect.

The rest of this article is organized as follows.Section 2 describes the theoretical foundations and assumptions.Section 3 discusses data and models.Section 4 provides an empirical analysis. Section 5 further explores the mechanisms of their impact.Section 6 concludes.

## 2. Theoretical foundations and assumptions

### 2.1 NDZ selection and urban green innovation

Unlike traditional innovation, green innovation places greater emphasis on achieving resource utilization efficiency improvement and reducing pollution through new technologies and ideas, while reaping economic benefits. Not only does it have the novelty and value characteristics of innovation in general, but it also has the characteristics of achieving resource conservation, environmental-friendly and sustainable development [21,22]. Green innovation is therefore also seen as a necessary means to generating sustainable advantage [23,24]. However, the positive externality, high risk, high investment and long cycle of green innovation have resulted in the rate of return of green innovation of enterprises being lower than that of

society, which reduces the incentive of green innovation of enterprises to a certain extent and triggers the dilemma of market failure. Industrial policy, as an important way for governments to compensate for market failures, can correct the positive externality of insufficient green innovation and bring the level of green innovation to the social optimum [25].

There are two types of industrial policy, one is the universal industrial policy which helps the market to function, the other is the selective policy which supports high-tech and strategic emerging industries. The selective policies can be traced back to the "catch-up theory", which uses policy support as a means of correcting market failures and aims to compress the development cycle of emerging industries and promote high-quality economic development. As an important selective industrial policy, the development zone plays an vital role in the process of China's economic development. In 1984, China established national-level economic and technological development zones in coastal cities open to the outside world for the first time. By 2018, China has established 552 NDZs and 1,991 provincial-level development zones. As China's development zones blossoming, they are gradually diversified in type. State-level development zones include economic and technological development zones (hereinafter referred to state-level economic and development zones), high-tech industrial development zones (hereinafter referred to state-level high-tech zones), export processing zones, comprehensive bonded zones, border cooperation zones, etc. Among them, state-level economic development zones and state-level high-tech zones hold an overwhelming advantage in economic contribution to the province. In 2019, 169 state-level high-tech zones and 218 state-level economic development zones in China achieved a total regional GDP of 22.9 trillion yuan, which equals to the sum of the GDP of Guangdong Province and Jiangsu Province in the same year, and therefore it is the key development zones mainly observed in this paper.

In the developing process of the development zones, the selection of NDZs in 2009 set off a upsurge, and the zones have officially entered the era of increasing quality and efficiency [26]. Differs from the establishment of development zones, the selection of NDZs is a process of selecting the best out of excellent, promoting provincial-level development zones to meet the criteria and being excellent in all aspects of development on the basis of the original [8]. After the upgrading of the Development Zone, it will obtain the signboard "National Brand", enjoy more tax incentives, and the location advantages of its jurisdiction will also be significantly improved. Specifically, the selection of NDZs is based on five selection criteria, including economic development, scientific and technological innovation, intensive and economical conservation, ecological protection and social responsibility. Thereafter, upgrade the provincial development zones that meet the standards to national development zones, in which scientific and technological innovation and ecological protection are important criteria for the selection of national development zones. Urban green innovation, as a combination of innovation and environmental benefits, is bound to become an effective tool for local governments to compete under a system of fiscal decentralization. On the one hand, local governments, under the dual pressure of pursuing regional economic growth targets and alleviating policy burdens, local governments hope to attract more investment and talent through the upgrading of development zones in their jurisdictions, so as to obtain more competitive advantages for themselves [27,28]. Therefore, in order to meet the requirements for the selection of NDZs, local governments will promote urban green innovation through various policies to increase the probability of upgrading the zones. On the other hand, after the provincial-level development zones are upgraded to NDZs, it will be more conducive to the exchange and collaboration of various innovation subjects within the zones, speeding up the flow and integration of innovation factors, which will in turn enhance the innovation capacity of the zones. Further, the increased innovation capacity of the park will become a driving force for technological progress and green growth in the city, reshaping the core competitiveness of the city and helping the city to

break its dependence on the original path of growth through the consumption of natural resources, thereby improving the overall level of green innovation in the city. In addition, local government competition is an act of strategic interaction between regions, and the efforts made by them to upgrade their provincial development zones prior to the selection of NDZs will also influence the strategic choices of their competitors and local governments in neighbouring regions. In the aftermath, the subjects in the park will accelerate the flow of knowledge and technology spillover through the market interaction with subjects in the surrounding areas, driving the level of green innovation in the surrounding cities through the linkage effect, with an obvious window radiation effect. Therefore, NDZs will have a catalytic effect on urban green innovation. In view of this, this paper proposes hypothesis H1.

H1:The selection of NDZs can promote green innovation in cities and has a window radiating effect.

## 2.2 Mechanisms of influence

As a merit-based policy, the selection of NDZs not only promotes local economic development, but also plays an important role in enhancing the level of green innovation in cities. In terms of the features of the NDZ selection policy, it is fundamentally different from the establishment of a NDZ. (1) Elevation of administrative level. NDZs, are mostly on a par with other municipal districts and have higher financial rights, while provincial-level development zones are merely a street or town under the jurisdiction of a district. (2) Promotion of policy incentives. While provincial-level development zones in China enjoy land tax exemptions, those selected as NDZs are able to enjoy national-level incentives such as VAT exemptions and access to more policy pilot opportunities. (3) Enhancement of advantages. Compared to provincial development zones, NDZs are more scarce, and a "National Permit" is more conducive to attracting capital and talent to the city. Under China's fiscal decentralization and performance appraisal system, NDZs are economic function areas for local governments to attract investment, and local governments have great power to upgrade provincial development zones. In order to meet the ecological protection and innovation conditions for the selection of NDZs, the most direct means for local governments is to strengthen environmental regulations, and this "competition for the top" can leave a catalytic impact on green innovation water in cities [29,30]. Moreover, after provincial development zones are selected for national development, the business environment of the zones will be further optimized. Therefore, better infrastructure and preferential policies will attract high-quality enterprises to enter, accelerating the flow of innovation factors and technology diffusion and improving the allocation efficiency of resources [31,32]. In addition, when a provincial-level development zone is successfully upgraded to a NDZ, the strength and scope of preferential policies will be further expanded, usually with generous tax incentives and government subsidies for enterprises in the zone, as well as priority treatment in terms of project approval, bank loans, labour organisation, and industrial land. A sort of policy supports can provide resources to support the high-investment and high-risk green innovation activities, which are conducive to the transformation of the city's growth pattern and the enhancement of green innovation. In summary, this paper argues that NDZ selection policies can promote green innovation in cities through the effects of environmental regulation, resource allocation and policy amplification.

**2.2.1 Environmental regulation effects.** There are two reasons why local governments are willing to compete for the selection of NDZs. One is that the decentralised fiscal system has stimulated local governments to put GDP above all else, and has encouraged them to develop their local economies [33,34]. The selection of NDZs provides local governments with opportunities for industrial clustering and investment attraction, and is an important grip for local

governments to expand the size of the local economy. The other, compared to provincial-level development zones, the industrial chain of NDZs is more complete. In order to solve the problem of future development bottlenecks and incomplete supporting industries for enterprises under their jurisdiction, local governments will actively strive for the opportunity to be selected as NDZs.

One of the conditions for the selection of NDZs is ecological environmental protection, which is evaluated in four aspects: implementation rate of environmental assessment for construction projects, emissions of sulfur dioxide per unit of GDP, emissions of chemical oxygen demand per unit of GDP and the incidence of pollution accidents in the development zones. The superior governments select and assess development zones under the jurisdiction of local governments according to their energy conservation and emission reduction targets, and upgrade provincial-level development zones that meet the requirements to NDZs. As "politicians", local governments will take corresponding measures in accordance with the selection criteria in order to obtain the "National Permit" and enhance the location advantage of their region. As a result, local governments often choose environmental regulation as a common tool to manage the environment in order to meet the requirements for selection as a NDZ. Due to the exit mechanism of national development zones, they are no longer "fixed status". Therefore, after the successful selection of a NDZ, this exit mechanism can also encourage local governments to pay continuous attention to the ecological environment of the zone in order to avoid being "removed from the list". After the upgrading of provincial development zones, local governments still be encouraged to engage in environmental regulation. As "economic agents", companies face strict environmental constraints before and after the selection of NDZ. In this case, companies shall make a choice based on their own situation [35]. Well-developed Enterprises will choose to transform and upgrade through green technology innovation, while those with poor development have to exit the development zone [36,37]. In this way, the development zone, after the survival of the fittest, the development zone will leave a number of enterprises with strong innovation ability and high awareness of environmental protection, which can in return promote the improvement of the green innovation level of the whole city. In view of this, this paper proposes hypothesis H2.

H2: NDZ selection promotes urban green innovation through environmental regulation effects.

**2.2.2 Resource allocation effects.**  According to the theory of spatial selection of resources, the costs and benefits of the located area, the spatial structure and the level of R&D influence the spatial selection of factors. High-end elements tend to flow into high-yield areas, and merit-based industrial policies undoubtedly provide a high-yield platform for those elements, especially when provincial development zones are upgraded to NDZs, which can provide new growth poles for the region [38]. The process of selecting the best out of excellent in this paper's NDZs will send a positive signal to enterprises, triggering high-technology enterprises to pile up and thus optimize the allocation of resources in terms of both capital and labour.

For one thing, the upgrading of the development zone to a "National Permit" is conducive to the city's FDI and industrial capital to move up the ladder, which, when combined with other high-end elements, can be helpful to provide new options for the city's economic development path and reduce its reliance on traditional natural resource extraction. Moreover, the concentration of capital elements brought about by NDZs also helps to "connect" green innovation, which not only greatly reduces the transaction costs within the urban green innovation system, but also solves the problem of insufficient capital input for green innovation.

Enterprises are able to obtain more convenient and larger scale funding support than before, which helps them to reduce the cost of using funds and enhance the efficiency of R&D, thus promoting a higher level of green innovation. As a result, a good green innovation atmosphere will further attract the inflow of capital, forming a virtuous cycle and ultimately bringing new drivers for the city to realize a more advanced and balanced development path and promote a higher level of green innovation in the city.

Secondly, as labour mobility is essentially a Pareto-improvement process, it will occur "foot voting" according to price, supply and demand, competition and other mechanisms. The successful selection of a NDZ will act as a magnetic pole to attract the workforce and help spread knowledge and experience among green innovation agents. On the one hand, the agglomeration of labour factors can also form a certain scale effect, contributing to form the phenomenon of "learning by doing" and "learning by learning" in the development zone. The interaction between subjects inside and outside the park can lead to the dissemination and spillover of knowledge, creating "linkages" inside and outside the park, and thus promoting green innovation in the city [39]. On the other hand, the dual incentive of the selection of NDZs and the policy of introducing talents can attract more high-end workforce to the city, which is a prerequisite for technological innovation and knowledge conversion, as well as the core driving force of green innovation development. It helps to increase the capital stock of urban labor force, which in turn enhances economic growth, improves the transformation of the economic growth model of the city, and promotes a higher level of green innovation in the city. Therefore, the selection of NDZs can optimize the allocation of resources by facilitating the flow of capital and labour factors, which in turn can promote green innovation in cities. In view of this, this paper proposes hypothesis H3.

H3:NDZ selection promotes urban green innovation through resource allocation effects.

**2.2.3 Policy amplification effect.** The upgrade of provincial development zones to NDZs has further expanded the scope of their policies and preferences. This paper argues that its policy amplification effect on urban green innovation is reflected in two aspects. One, as national and provincial development zones are at different levels, their approval bodies and management units may also differ, resulting in different government resources being available to enterprises in the domain. After the selected as a NDZ, its policy advantages and development strategies are gradually aligned with the national level, and it has more independent authority to play a "leading role" in regional development [8]. In particular, upgrading to a NDZ will be given greater autonomy to reform, with more streamlined approval procedures and easier access to state support for major projects. For example, in 2013, Shanghai Zhangjiang Hi-tech Zone delegated 13 categories and 20 items of administrative approval rights, which include land grant, approval of foreign-invested enterprises, environmental impact assessment and preventive health audit, and some foreign-invested enterprises can achieve in-park approval. The 2019 Opinions on Promoting Innovation and Enhancement in State-level Economic and Technological Development Zones to Create a New Highland of Reform and Opening Up also emphasises streamlining the procedures for accessing investment projects in accordance with the law, simplifying approval procedures, decentralising provincial and municipal economic management approval authority, and implementing a new model of construction before inspection management. This progressive increasing in policy support will help to reduce the institutional transaction costs of enterprises in terms of compliance, distribution of intellectual property rights and interests, and transformation of innovation results [40]. At the same time, following the major rectification of development zones, the central government requires the construction of state-level development zones to gradually shift to a new model of industrial

optimisation, quality and efficiency enhancement, innovation-driven and stable development. Guided by this development strategy, enterprises will also focus more on organic growth and green innovation levels, so as to drive green innovation of the whole city.

Secondly, after selected as the NDZs, it can enjoy greater tax incentives and government subsidies, which is an important factor in promoting green innovation in cities. The tax incentives in NDZs, for example, can effectively reduce the R&D costs of enterprises and compensate for some of the asset losses incurred in the process of green innovation activities, which undoubtedly alleviates the financing constraints of enterprises to a certain extent and enhances their incentive for green innovation [41,42]. At the same time, tax incentives can also reduce R&D costs and uncertainty for companies, and effectively disperse the risk of green R&D activities [43]. In addition, unlike provincial development zones, NDZs are eligible to issue long-term bonds with banks, set up venture capital funds and have priority rights in bank loans. Moreover, the upgrading of development zones has a stronger "recognition effect", especially for enterprises that are of great significance to the national development strategies and have already received policy support, which can send more positive signals to the outside world and thus receive more financial support for green innovation and promote green innovation in the city as a whole [44]. In view of this, this paper proposes hypothesis H4.

H4: NDZ selection promotes urban green innovation through policy amplification.

## 3. Study design

### 3.1 Data sources and processing

This paper uses Chinese prefecture-level city-level data from 2000-2018. To avoid the effect of outliers, the author excludes samples from Xinjiang and parts of Tibet. The data on development zones are obtained from the *Catalogue of China's Development Zones Review and Announcement (2006)* and *Catalogue of China's Development Zones Review and Announcement (2018 Edition)*, and the data on patents are obtained from the State Intellectual Property Office. Other city-level data are from the *China City Statistical Yearbook*, *China Regional Economic Statistical Yearbook* and *China Statistical Yearbook*.

### 3.2 Model specification

The selection of NDZs can be regarded as a "quasi-natural experiment" and the existing literature mainly adopt the method of DID [8]. However, the theoretical part, this paper reveals that the selection of NDZs can not only have an impact on the green innovation of the city, but also have a window radiation effect on the green innovation of surrounding cities. Using the traditional DID method cannot explain the spatial spillover effect brought about by the selection of NDZs, and it is easy to bring about multi-collinearity problems between dummy variables and cross-terms. Therefore, based on the traditional DID model, this paper proposes that the spatial durbin model can not only examine the spatial lag and impact of the spatial error term, but also examine the external influence of the spatial lag term of the independent variable. Thus a DID space durbin model is set as follows:

$$UGI_{i,t} = \alpha + \beta_1 \sum_{i=1}^{n} (\zeta_{i,j} \otimes w_{i,j}) UGI_{i,t} + \beta_2 NDZS_{i,t} + \beta_3 (\zeta_{i,j} \otimes w_{i,j}) NDZS_{i,t} + \beta_4 \text{control}_{i,t}$$
$$+ D1_{i,t} + D2_{i,t} + \varepsilon_{i,t} \tag{1}$$

where $i$ represents the city, $t$ represents the time, and $UGI_{i,t}$ represents urban green innovation. $NDZS_{i,t}$ is the combined dummy variable selected by NDZs, which is composed of the product

of the data group and the centralized value of the dummy variable selected by NDZs, namely $NDZS_{i,t} = \left(NDZS_{i,t}^{(1)} - \overline{NDZS}^{(1)}\right) \times \left(NDZS_{i,t}^{(2)} - NDZS^{(2)}\right)$. $NDZS_{i,t}^{(1)}$ are dummy variables of data group, which takes the value of 1 when the data point is in the treatment group, and 0 otherwise; $NDZS_{i,t}^{(2)}$ is a dummy variable selected for NDZs selection, and the years upon and after the NDZ selection take the value of 1 for each year, and 0 otherwise. $\zeta_{i,j} \otimes w_{i,j}$ is the endogenous space-time weight matrix, where $\zeta_{i,j}$ and $w_{i,j}$ are the time weight matrix and the space weight matrix, respectively. Referring to existing researchers, this paper utilizes both geographical and economic spatial weighting matrices [45,46]. Among them, the geospatial weight matrix represents the geographic spatial relationship between city $i$ and city $j$, which is calculated as the reciprocal of the distance between cities, and the economic spatial weight matrix represents the spatial relationship between city $i$ and city $j$ on the level of economic development. In this paper, the spatial weight matrix is set according to the similarity of urban per capita GDP. $D1_{i,t}$ and $D2_{i,t}$ are urban fixed effects and year fixed effects, respectively, and $\varepsilon_{i,t}$ are random error terms.

## 3.3 Model specification

Urban Green Innovation (*UGI*): The State Intellectual Property Office provided relevant information such as patent application number, application date, publication number, publication date, patent name, abstract, classification number, applicant, and inventor. Compared with design patents, utility model patents and invention patents have higher technical content. Thus this paper chooses them as the research objects of innovation. Furthermore, according to the "IPC Green Inventory" published by the World Intellectual Property Organization (WIPO), this paper matches and identifies green patents through patent classification numbers and sum up them to the city level. Use the total number of green patents (the sum of green invention patents and green utility patents), green invention patents and green utility patents to measure the level of urban green innovation, expressed as *UGI*, *UGI1* and *UGI2* respectively, adding 1 and taking logarithm.

Selection of NDZs: This paper focuses on the upgrade of provincial-level development zones to NDZs, and selects cities without NDZs before 2006 according to the *China Development Zone Audit Announcement Catalogue (2006)*. According to the *China Development Zone Review Announcement Catalogue (2018 Edition)*, we further sort out the cities that had provincial-level development zones selected as NDZs from 2009 to 2016, and set the value to 1, and 0 otherwise.

Environmental regulatory effects: This paper applys the intensity of environmental regulation (*IER*) to measure the environmental regulatory effect brought about by the selection of NDZs. Referring to the existing practices, the environment is measured by constructing a comprehensive index for the discharge of waste water, $SO_2$, and smoke/dust. Regulatory strength [47]. First, according to the three types of pollutants discharged per unit economic output of each city, linear standardization is carried out. $PD_{ik}^{S} = \frac{PD_{ik} - \min(PD_k)}{\max(PD_k) - \min(PD_k)}$, where $PD_{ik}$ is the emission of pollutants per unit output value of city $i$, $PD_{ik}^{S}$ is the value of the index after standardized treatment, $\max(PD_k)$ and $\min(PD_k)$ respectively represent the maximum and minimum emissions of various pollutants in city $i$. Then, set the adjustment parameters to reflect the differences in pollution in each city. The adjustment parameters are $W_k = PD_{ik}/\overline{PD_{ik}}$. Finally, the intensity of environmental regulations in each city is calculated as: $IER_i = \sum_1^k W_k PD_{ik}^{S}$.

Resource allocation effect: This paper reflects the effect of resource allocation by measuring the reverse index of the degree of factor market distortion. Here, this paper focuses on two aspects: capital market factor distortion and labor market factor distortion. Drawing on

existing literature, the degree of distortion of capital market factors and labor market factors can be obtained by dividing the marginal output of labor and capital by its price. Then take logarithm to get the resource allocation efficiency of the capital market and labor market, denoted as *CU* and *LU* respectively. The larger the *CU* value, the smaller the distortion of the capital market, and the higher the efficiency of capital market resource allocation; similarly, the larger the *LU* value, the smaller the distortion of the labor market, and the higher the efficiency of labor market resource allocation [48].

Policy Amplification Effect: Considering that the biggest difference between NDZs and provincial-level development zones lies in tax incentives, this paper uses tax incentives (*policy*), the taxation of industries above designated size at city level to express the policy amplification effect.

Control variables: The control variables selected in this paper include: (1)Urban economic development level (*Economy*). This paper uses per capita GDP as an indicator to measure the level of urban economic development, taking the logarithm. (2)For the level of opening-up (*Open*), this paper takes the proportion of FDI in GDP as an alternative index (3)The level of infrastructure construction *(inf)* is measured by the highway mileage (logarithm) of prefecture level cities in this paper. (4)Industrial structure (*lngdp3*). This paper measures the industrial structure by the proportion of the tertiary industry in GDP. (5)Fiscal expenditure bias (*fs*). This paper uses the proportion of local fiscal general budgetary expenditure after deducting livelihood expenditure. (6)Human capital (*hc*). This paper measures the number of college students per 10,000 students (logarithm).

### 3.4 Summary statistics

The descriptive statistics for the variables in this article are shown in Table 1.

## 4. Empirical results

### 4.1 Baseline regression results

Due to the spatial correlation, the level of green innovation in each city is influenced not only by factors related to the province, but also by factors related to other provinces. In view of this, this paper constructs a global Moran I index to test whether there is a correlation between the variables. It is found that the total number of green patents, green invention patents and green

**Table 1. Summary statistics.**

| Variable | Notation | Obs | Mean | Std. dev. | Min | Max |
|---|---|---|---|---|---|---|
| Total of green patents | *UGI* | 5415 | 3.474 | 1.937 | 0 | 9.962 |
| Green invention patents | *UGI1* | 5415 | 2.627 | 1.906 | 0 | 9.543 |
| Green utility patents | *UGI2* | 5415 | 2.993 | 1.829 | 0 | 8.890 |
| NDZ selection | *NDZS* | 5415 | 0.442 | 0.497 | 0 | 1 |
| Environmental regulation | *IER* | 5415 | 0.245 | 0.757 | 0 | 24.541 |
| Urban economic level | *Economy* | 5415 | 9.980 | 0.947 | 4.595 | 15.672 |
| Opening-up level | *Open* | 5415 | 0.023 | 0.031 | 0 | 0.775 |
| Infrastructure level | *Inf* | 5415 | 9.351 | 1.187 | 5.143 | 12.926 |
| Industrial structure | *lngdp3* | 5415 | 0.379 | 0.095 | 0.085 | 0.853 |
| Fiscal expenditure level | *fs* | 5415 | 0.611 | 0.083 | 0.023 | 0.965 |
| Human capital | *hc* | 5415 | 10.051 | 1.518 | 3.401 | 13.911 |

Notes: Based on Stata.

utility patents in cities are all significantly positive in the Moran I index from 2000 to 2018, showing a strong spatial correlation. A spatial econometric model can be used to further empirical test.

To test the relationship between the selection of NDZs and urban green innovation, this paper makes a regression based on model (1). All regression results include control variables with time fixed effects and city fixed effects, and the regression results are reported in Table 2. It can be found that there is a significant positive relationship between the selection of NDZs and the total number of green patents (*UGI*), green invention patents (*UGI*1) and green utility model patents (*UGI*2) in cities. This paper argues that the reason for this is that the local government, in order to meet the criteria for upgrading a development zone, will promote urban green innovation through various policies to increase the probability of upgrading the zone. The location advantages that arise when a development zone is upgraded to a national level can also provide a better platform for green innovation, which in turn helps to enhance the level of green innovation in the city. In addition, Table 2 shows that the spatial spillover effect of the selection of NDZs is significantly positive, indicating that the selection of NDZs has a significant spatial radiation effect, which can not only promote green innovation in the city, but also drive green innovation in neighbouring cities, realizing a two-way flow of factors and thus forming an obvious positive spatial spillover effect. In view of this, H1 obtained evidence.

## 4.2 Heterogeneity tests

**4.2.1 Old vs. young industrial cities.** Usually, old industrial cities are heavily polluted due to their excessive dependence on industrialization. In order to explore whether the flow of factors and local government behaviors caused by the opening of the Selection of NDZs will have a different impact on the green innovation of old industrial bases and young industrial cities, this paper divides the sample into old industrial bases and young industrial cities according to the *National Plan for Adjustment and Reconstruction of Old Industrial Bases (2013-2022)* for further testing. From the regression results in Table 3, the Selection of NDZs selection in old industrial cities has no significant impact on the total number of green patents, green invention patents, and green utility patents. This paper holds the view that the reason for this is that old industrial cities have always had the characteristics of high energy consumption, and the production method has serious path dependence, and it is difficult to achieve the improvement of the urban green innovation level in the short term. On the one hand, due to the limitations of its own conditions, motivation to participate in the selection of NDZs is insufficient; On the other hand, even if it is selected as a National Development Zone, it is

**Table 2. Baseline regression results.**

| Variable | *UGI* | *UGI*1 | | | | *UGI*2 | | | |
|---|---|---|---|---|---|---|---|---|---|
| | spatial durbin model | spatial Durbin model | spatial decomposition | | | spatial durbin model | spatial decomposition | | |
| | | | direct | indirect | total effect | | direct | indirect | total effect |
| *NDZS* | 0.097*** (0.050) | 0.115** (0.062) | 0.172*** (0.063) | 1.369*** (0.301) | 1.541*** (0.311) | 0.149*** (0.046) | 0.201*** (0.048) | 1.289*** (0.192) | 1.489*** (0.245) |
| *W×NDZS* | 0.238** (0.147) | 0.501*** (0.140) | - | - | - | 0.541*** (0.096) | - | - | - |
| control variable | YES | YES | YES | YES | YES | YES | YES | YES | YES |
| adjusted R² | 0.835 | 0.791 | | | | 0.836 | | | |
| obs | 5 415 | 5 415 | 5 415 | 5 415 | 5 415 | 5 415 | 5 415 | 5 415 | 5 415 |

Notes: Due to space limitation, the control variable coefficients are not reported. Numbers in brackets are robustness standard errors;

***, ** and * indicate significance levels at 1%, 5% and 10%, respectively. Control variables, time FE and city FE are all controlled. The same below.

**Table 3. Heterogeneity test: Old vs. young industrial cities.**

| Variable | Old industrial cities | | | Young industrial cities | | |
|---|---|---|---|---|---|---|
| | **(1)** **UGI** | **(2)** **UGI1** | **(3)** **UGI2** | **(4)** **UGI** | **(5)** **UGI1** | **(6)** **UGI2** |
| NDZS | -0.021(0.054) | 0.025 (0.068) | -0.015 (0.054) | 0.165** (0.067) | 0.096* (0.041) | 0.211*** (0.063) |
| W×NDZS | 0.386** (0.178) | 0.406 (0.249) | 0.342** (0.149) | 0.499*** (0.167) | 0.422** (0.214) | 0.684*** (0.159) |
| control variable | YES | YES | YES | YES | YES | YES |
| adjusted $R^2$ | 0.828 | 0.787 | 0.807 | 0.833 | 0.789 | 0.818 |
| obs | 2166 | 2166 | 2166 | 3249 | 3249 | 3249 |

difficult to offset the negative impact of the old industrial city's own environment on green innovation. For young industrial cities, the main effect coefficient and spatial spillover coefficient of the national development zone selection are positive, indicating that the national development zone selection can significantly improve the green innovation level of its city and surrounding cities.

**4.2.2 City size.**   The larger the city, the more it will be "favoured" by the central government, thus the more it can enjoy economic priority and overlap with NDZs to influence urban green innovation. Therefore, this paper classifies cities into large, medium-size and small cities according to the *Notice on City Size Classification* issued by the State Council of China. The regression results are reported in Table 4. It can be found that in large and medium-sized cities, the NDZ selection policy can have a positive impact on urban green innovation. It is mainly because the suitable effect of the NDZ selection policy relies on the economic base of the region, and large and medium-sized cities tend to be strong and have abundant resources to escort the improvement of the city's green innovation level. In contrast, the NDZ selection policy for small cities has no significant impact on urban green innovation, which probably due to their weak foundation of economic development. It will take times for the effect of green innovation in provincial development zones, even if they are upgraded to NDZs.

**4.2.3 Different levels of transportation infrastructure.**   The transport infrastructure, as the necessary environment of the development zone, not only improves the accessibility of transportation, but also facilitates the spatial flow of factors and optimises the allocation of resources. It can also enhance the location advantage and attract more inflow of enterprises and capital, which is of great significance to the development of the development zone. Scholars have also pointed out that the opening of highways and the construction of railways (or high-speed trains) can help reduce trade costs and improve economic efficiency [49,50].

**Table 4. Results of the city size heterogeneity test.**

| Variable | Large cities | | Medium-sized cities | | Small Cities | |
|---|---|---|---|---|---|---|
| | **(1)** **UGI** | **(2)** **UGI1** | **(3)** **UGI** | **(4)** **UGI1** | **(5)** **UGI** | **(6)** **UGI1** |
| NDZS | 0.115*** (0.024) | 0.275** (0.111) | 0.175** (0.077) | 0.226** (0.089) | 0.018 (0.060) | 0.017 (0.073) |
| W×NDZS | 0.678*** (0.065) | 0.543*** (0.154) | 0.570** (0.267) | 0.516*** (0.153) | 0.072 (0.174) | 0.035 (0.214) |
| control variable | YES | YES | YES | YES | YES | YES |
| adjusted $R^2$ | 0.819 | 0.778 | 0.827 | 0.785 | 0.824 | 0.778 |
| obs | 266 | 266 | 1672 | 1672 | 3477 | 3477 |

Table 5. Heterogeneity test: Convenience vs. inconvenience transportation.

| Variable | Convenience transportation group | | | Inconvenience transportation group | | |
|---|---|---|---|---|---|---|
| | (1) UGI | (2) UGI1 | (3) UGI2 | (4) UGI | (5) UGI1 | (6) UGI2 |
| NDZS | 0.153*** (0.055) | 0.204*** (0.067) | 0.832*** (0.051) | -0.142 (0.111) | -0.279* (0.138) | -0.044(0.108) |
| W×NDZS | 0.488*** (0.149) | 0.402** (0.185) | 0.595*** (0.137) | -1.005* (0.511) | -0.971* (0.345) | -0.901* (0.308) |
| control variable | YES | YES | YES | YES | YES | YES |
| adjusted R$^2$ | 0.848 | 0.801 | 0.832 | 0.821 | 0.778 | 0.799 |
| obs | 4142 | 4142 | 4142 | 1273 | 1273 | 1273 |

However, the level of development of cities varies, so does the level of transport infrastructure. Considering that the construction of high speed railways is more conducive to the economic level, this article focuses on the construction of high speed railways in cities. In order to investigate whether the different levels of transport infrastructure in cities belonging to the development zones will affect the implementation of the policy, this paper groups cities according to whether they have a high-speed railway station or not, defining cities with a high-speed railway station as the high accessibility group and cities without a high-speed railway station as the non-accessibility group. The regression results are reported in Table 5. It can be found that in cities cities with convenient transportation, the NDZ selection policy plays a significant role in promoting urban green innovation, while for cities without advanced transportation infrastructure, the NDZ selection policy has no significant impact on urban green innovation, so it is difficult to play its policy role. The better the transport infrastructure of the area, the more conducive it is to the flow of factors and economic development, which can provide a solid foundation for the NDZ selection policy and facilitate its positive effect on urban green innovation.

**4.2.4 Level of marketability.** The fact that developed regions with higher market-oriented institutional environment have a higher level of economic culture than regions with a lower level of marketisation which inevitably leads to a heterogeneous impact of NDZ selection policies on urban green innovation. This paper defines regions with a marketization index at or above the median as high marketization level cities and otherwise as low marketization level cities, based on the *2018 China Regional Marketization Index Report* co-sponsored by the China National Economic Research Institute and Social Sciences Academic Press(CHINA). The regression results are reported in Table 6. It shows that the green innovation effect of the NDZ selection policy is significant high market-oriented cities and insignificant in low

Table 6. Heterogeneity test: Highly marketable vs. low-market.

| Variable | Highly marketable cities | | | Low-market cities | | |
|---|---|---|---|---|---|---|
| | (1) UGI | (2) UGI1 | (3) UGI2 | (4) UGI | (5) UGI1 | (6) UGI2 |
| NDZS | 0.129** (0.064) | 0.157** (0.077) | 0.155** (0.061) | 0.018 (0.069) | 0.009 (0.084) | 0.085 (0.065) |
| W×NDZS | 0.362*** (0.162) | 0.531*** (0.159) | 0.274* (0.148) | 0.168 (0.235) | 0.184 (0.307) | 0.322 (0.214) |
| control variable | YES | YES | YES | YES | YES | YES |
| adjusted R$^2$ | 0.831 | 0.785 | 0.813 | 0.824 | 0.782 | 0.808 |
| obs | 2812 | 2812 | 2812 | 2603 | 2603 | 2603 |

**Table 7. Heterogeneity test: High government governance vs.low government governance.**

| Variable | High government governance | | | Low government governance | | |
|---|---|---|---|---|---|---|
| | (1) UGI | (2) UGI1 | (3) UGI2 | (4) UGI | (5) UGI1 | (6) UGI2 |
| NDZS | 0.132* (0.078) | 0.103*** (0.012) | 0.190** (0.073) | 0.038 (0.060) | 0.058 (0.074) | 0.078 (0.055) |
| W×NDZS | 0.797*** (0.238) | 0.769*** (0.274) | 1.029*** (0.233) | -0.354 (0.575) | -0.432 (0.605) | -0.361 (0.454) |
| control variable | YES | YES | YES | YES | YES | YES |
| adjusted $R^2$ | 0.844 | 0.797 | 0.832 | 0.808 | 0.763 | 0.788 |
| obs | 1900 | 1900 | 1900 | 3515 | 3515 | 3515 |

market-oriented cities. It shows that the upgrading of provincial development zones to NDZ gives them more independent authority and is more capable of creating synergies with market-based forces, which in turn promotes a higher level of green innovation in the city. This empirical result is also consistent with the reality in China: the top-ranked national economic development zones and national high-tech zones are mostly concentrated in areas with a high level of marketisation, such as the eastern coast. In cities with backward marketisation, where economic operations and resource management are mostly interfered by the intervention of local governments, the role of NDZs will be limited, making it difficult for them to play an effective role in promoting green innovation in cities.

**4.2.5 Level of government governance.** The effectiveness of the implementation of the NDZ selection policy is inextricably linked to the level of government governance. To test whether the impact of the policy on urban green innovation varies according to the level of government governance, this paper uses the *2016 China Local Government Management Effectiveness Ranking*, jointly relished by Management Insights and the Institute of Government Management of Beijing Normal University, to classify provincial government efficiency according to intermediate rankings. The government efficiency ranking in the top 50% is defined as a high level of government governance, otherwise it is a low level of government governance. The results of the test are reported in Table 7. It shows that when the government efficiency ranks is in the top 50%, the NDZ selection policy can significantly promote urban green innovation. This paper argues that the reason for this is that an "active" government can not only provide an efficient and convenient business environment and opportunities for communication between innovation agents, but can also effectively combine market forces to provide the right guidance to enterprises through the formulation of policies and green development strategies, thus enabling the city closer to an intensive development model and improve the overall level of green innovation.

**4.2.6 Level of development of the development area.** The better the level of economic development the better the implementation of the policy effect. Development zones with a high level of economic development have a unique development advantage, which may be further amplified by the NDZ selection policy, and in turn, leave a differential infulence on the level of green innovation in different cities. Thefore, this paper uses the average annual industrial value added density to measure the development level of development zones and divides the sample into high-level development zone and low-level development zone. The regression results are reported in Table 8. It shows that development zones with high economic development are more effective in enhancing the city's green innovation level after upgrading to NDZs, while those, with low economic development, the green innovation effect of the national selection policy can not be effectively brought into play.

**Table 8. Heterogeneity test: High level of development vs.low level of development.**

| Variable | High Development Zone | | | Low Development Zone | | |
|---|---|---|---|---|---|---|
| | (1) UGI | (2) UGI1 | (3) UGI2 | (4) UGI | (5) UGI1 | (6) UGI2 |
| NDZS | 0.184*** (0.051) | 0.127** (0.054) | 0.162*** (0.024) | 0.127 (0.368) | 0.072 (0.122) | 0.109 (0.089) |
| W×NDZS | 0.238* (0.144) | 0.361*** (0.164) | 0.532*** (0.158) | 0.168* (0.087) | 0.061 (0.337) | 0.145 (0.231) |
| control variable | YES | YES | YES | YES | YES | YES |
| adjusted $R^2$ | 0.816 | 0.834 | 0.787 | 0.804 | 0.778 | 0.821 |
| obs | 1558 | 1558 | 1558 | 3857 | 3857 | 3857 |

### 4.3 Robustness check

**4.3.1 Parallel trend test.** The key premise of applying the DID is that the treatment group and the control group have the same change trend before the implementation of the policy, and therefore, this paper treats the event of NDZ selection as forward and backward three phases respectively to test whether it meets the parallel trend hypothesis. The test results are reported in Fig 1. The vertical ordinate is the city, which demonstrates that the averages of the first three phases of the NDZ selection are not significant, while the averages of the three phases after the NDZ selection have a significant positive impact. And with time goes by, the gaps between the selected and non-selected cities widens, indicating that the conclusions of this paper are stable.

**4.3.2 Eliminating metropolises.** Compared with other cities, the economic development level of the four metropolises directly under the central government (Beijing, Shanghai, Tianjin, and Chongqing) is far ahead. In addition, whether the city participates in the NDZ

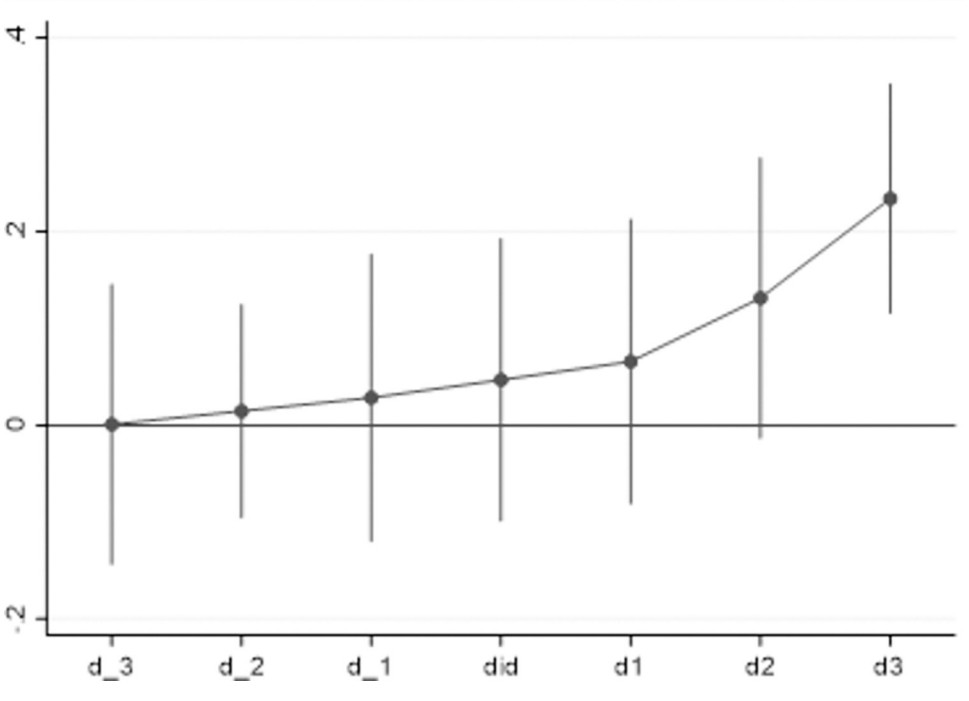

**Fig 1. Parallel trend test.**

**Table 9. Robustness check: Removing four metropolises.**

| Variable | *UGI* | *UGI1* | | | | *UGI2* | | | |
|---|---|---|---|---|---|---|---|---|---|
| | spatial durbin model | spatial durbin model | spatial decomposition | | | spatial durbin model | spatial decomposition | | |
| | | | direct | indirect | total effect | | direct | indirect | total effect |
| *NDZS* | 0.097*** (0.025) | 0.119** (0.061) | 0.176*** (0.063) | 1.387*** (0.305) | 1.563*** (0.316) | 0.160*** (0.046) | 0.225*** (0.049) | 1.589*** (0.221) | 1.813*** (0.226) |
| *W×NDZS* | 0.237*** (0.073) | 0.501*** (0.140) | - | - | - | 0.597*** (0.095) | - | - | - |
| control variable | YES | YES | YES | YES | YES | YES | YES | YES | YES |
| adjusted R$^2$ | 0.832 | 0.789 | | | | 0.829 | | | |
| obs | 5339 | 5339 | 5339 | 5339 | 5339 | 5339 | 5339 | 5339 | 5339 |

selection is non-random. Therefore, the existence of outliers may affect the regression results. Based on the above, this paper excludes the samples of the four municipalities and then performs the test. The regression results are reported in Table 9, which shows that the conclusions of this paper are still stable.

**4.3.3 The lagging effect of NDZ.** Given that compared with other productive activities, innovation takes more time, this paper treats the NDZ selection policy as a lagged period to reflect the lagged effect on urban green innovation. The regression results are reported in Table 10. It suggests that the NDZ selection policy has a positive impact on urban green innovation and has a significant spatial spillover effect, indicating that the conclusions of this paper are stable.

**4.3.4 Placebo test.** In order to further verify that the NDZ selection policy is not affected by the omitted variables, this paper refers to the method of Alder, randomly assigns the number of NDZ selections in each year to each city, and repeats this process 1000 times, as shown in Fig 2. The kernel density map of the NDZ selection coefficient obtained by this method, the average value of the coefficient after random processing is 0.0081, which is very close to 0 compared with the baseline result, which verifies that the NDZ selection has improved the level of urban green innovation [8].

## 5. Mechanism test and analysis

### 5.1 Mechanism test model

To discuss the mechanism of the impact of NDZ selection on urban green innovation, the author first test the influence of NDZ selection on the intermediary variables, and then by introducing the interaction term between the NDZ and the intermediate variables, test

**Table 10. Lagging effects of NDZ.**

| Variable | *UGI* | *UGI1* | | | | *UGI2* | | | |
|---|---|---|---|---|---|---|---|---|---|
| | spatial durbin model | spatial durbin model | spatial decomposition | | | spatial durbin model | spatial decomposition | | |
| | | | direct | indirect | total effect | | direct | indirect | total effect |
| *NDZS* | 0.091*** (0.026) | 0.109*** (0.031) | 0.130*** (0.033) | 0.523*** (0.171) | 0.653*** (0.186) | 0.142*** (0.026) | 0.172*** (0.028) | 0.775*** (0.135) | 0.946*** (0.146) |
| *W×NDZS* | 0.247*** (0.075) | 0.211** (0.089) | - | - | - | 0.359*** (0.076) | - | - | - |
| control variable | YES | YES | YES | YES | YES | YES | YES | YES | YES |
| adjusted R$^2$ | 0.837 | 0.792 | | | | 0.822 | | | |
| obs | 5415 | 5415 | | | | 5415 | | | |

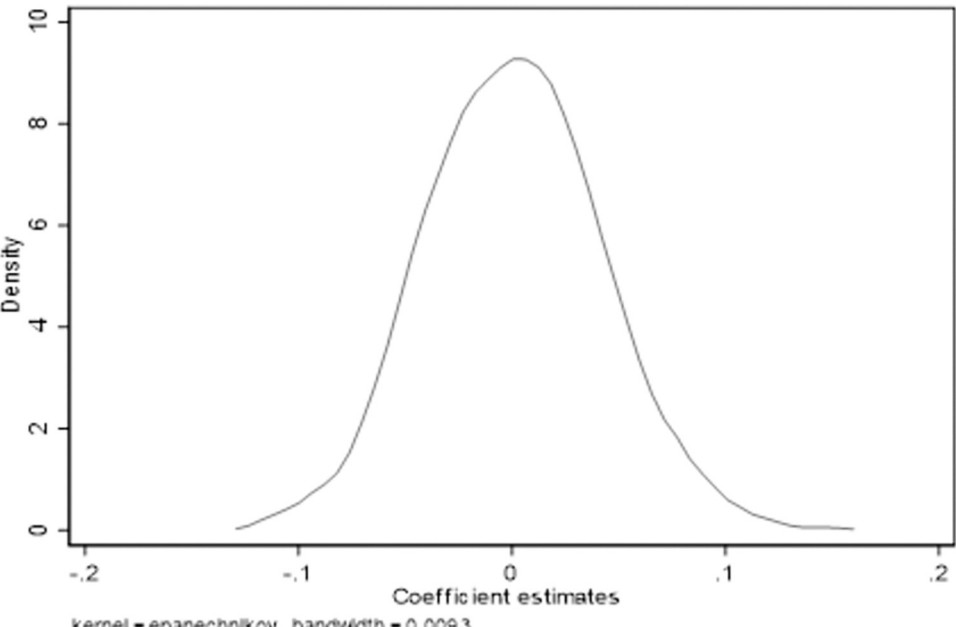

**Fig 2. Placebo test.**

whether the intermediary variables play an intermediary role in the impact of the NDZ selection on urban green innovation. The model is set as follows:

$$Z_{i,t} = \mu + \eta_1 \sum_{i=1}^{n} (\zeta_{i,j} \otimes w_{i,j}) Z_{i,t} + \eta_2 NDZS_{i,t} + \eta_3 (\zeta_{i,j} \otimes w_{i,j}) HNDZS_{i,t_{i,t}} + \eta_4 \text{control}_{i,t} + D1_{i,t} + D2_{i,t} + \varepsilon_{i,t} \tag{2}$$

$$UGI_{i,t} = \theta + \rho_1 \sum_{i=1}^{n} (\zeta_{i,j} \otimes w_{i,j}) UGI_{i,t} + \rho_2 NDZS_{i,t} \times Z_{i,t} + \rho_3 (\zeta_{i,j} \otimes w_{i,j}) NDZS_{i,t} \times Z_{i,t} + \rho_4 \text{control}_{i,t} + D1_{i,t} + D2_{i,t} + \varepsilon_{i,t} \tag{3}$$

Among them, the intermediary variables include environmental regulation (*IER*), capital and labor market allocation efficiency (*CU* and *LU*), and tax incentives (*policy*). Other items have the same meaning as model (1). The coefficients represent the influence of the NDZ selection on the intermediary variables and the spatial spillover effect respectively. This part focuses on the interaction coefficient sum, which reflects whether the NDZ selection has an impact on urban green innovation through environmental regulation, resource allocation, and policy amplification effect.

## 5.2 Mechanism test results

**5.2.1 The effect of environmental regulations.** In the theoretical part, this paper puts forward the hypothesis that the NDZ selection will have an impact on urban green innovation through environmental regulatory effects. This section verifies this, and the regression results are reported in Table 11. From the column (2) of Table 11, it reveals that the NDZ policy has indeed produced environmental regulation effects, and the coefficient on the interactive items between environmental regulations and NDZ selection is significantly positive at the level of 1%, indicating that the NDZ policy can motivate local governments to improve environmental

**Table 11. Mechanism test-environmental regulatory effect.**

| Variable | Environmental Regulatory Effect | | |
|---|---|---|---|
| | (1) UGI | (2) IER | (3) UGI |
| NDZS | 0.097*** (0.050) | 0.067** (0.029) | |
| W×NDZS | 0.238** (0.147) | 0.287** (0.132) | |
| NDZS×IER | | | 0.159*** (0.047) |
| W×NDZS×IER | | | 0.387*** (0.131) |
| control variable | YES | YES | YES |
| adjusted R$^2$ | 0.835 | 0.719 | 0.832 |
| obs | 5 415 | 5 415 | 5 415 |

regulations and restrict local pollution emissions, thereby enhancing the level of urban green innovation. In view of this, H2 is proven.

**5.2.2 Resource allocation effect.** In the theoretical part, this paper puts forward the hypothesis that the NDZ policy will have an impact on urban green innovation through the effect of resource allocation. This section verifies this, and the regression results are reported in Table 12. From the columns (2) and (5) of Table 12, it demonstrates that the selection of NDZs can indeed optimize the allocation of capital and labor, and whether it is capital allocation, and the coefficients of the interaction terms between labor allocation and the selection of NDZs are both significantly positive, indicating that the NDZ policy can optimize local resource allocation by attracting capital and labor factors, reshape the core competitiveness of cities, and achieve high-quality development. In particular, the successful selection of NDZs can provide a better platform for high-end talents, which will attract a large inflow of innovative talents, which in turn will generate knowledge spillover effects and promote urban green innovation. In view of this, H3 is proven.

**Table 12. Mechanism test-resource allocation effect.**

| Variable | Capital allocation | | | Labor allocation | | |
|---|---|---|---|---|---|---|
| | (1) UGI | (2) CU | (3) UGI | (4) UGI | (5) LU | (6) UGI |
| NDZS | 0.097*** (0.050) | 0.095* (0.054) | | 0.097*** (0.050) | 0.174*** (0.003) | |
| W×NDZS | 0.238** (0.147) | 0.767** (0.012) | | 0.097*** (0.050) | 0.093** (0.052) | |
| NDZS×CU | | | 0.008* (0.004) | | | |
| W×NDZS×CU | | | 0.037*** (0.008) | | | |
| LU×NDZS | | | | | | 0.371*** (0.041) |
| W×LU×NDZS | | | | | | 0.969*** (0.211) |
| control variable | YES | YES | YES | YES | YES | YES |
| adjusted R$^2$ | 0.835 | 0.664 | 0.856 | 0.821 | 0.329 | 0.7847 |
| obs | 5 415 | 5 415 | 5 415 | 5 415 | 5 415 | 5 415 |

**Table 13. Mechanism test-policy amplification effect.**

| Variable | Policy Amplification Effect | | |
|---|---|---|---|
| | (1)<br>*UGI* | (2)<br>*Policy* | (3)<br>*UGI* |
| *NDZS* | 0.097***<br>(0.050) | 0.161***<br>(0.017) | |
| *W×NDZS* | 0.097***<br>(0.050) | 0.129***<br>(0.037) | |
| *Policy×NDZS* | | | 0.124***<br>(0.041) |
| *W×Policy×NDZS* | | | 0.211***<br>(0.101) |
| control variable | YES | YES | YES |
| adjusted $R^2$ | 0.821 | 0.784 | 0.798 |
| obs | 5 415 | 5 415 | 415 |

**5.2.2 Policy amplification effect.**   In the theoretical part, this paper puts forward the hypothesis that the selection of NDZs will amplify the effect through policies, and then have an impact on urban green innovation. This section verifies this, and the regression results are reported in Table 13. From the column (2) of Table 13, it indicates that the selection of NDZs does have a policy amplification effect, and the interaction coefficients of tax incentives and the selection of NDZs significantly positive at the 1% level, indicating that the successful selection of NDZs will bring more supporting policies, and enterprises in the region are more likely to be favored by national development strategies. In other words, the selection of NDZs will induce more policies to amplify the positive effect, thereby promoting urban green innovation. In view of this, H4 is proven.

## 6. Conclusion and policy implications

Based on the data on Chinese cities from 2000 to 2018, this paper integrates the innovation and environmental effects of selecting NDZs using a double-difference spatial durbin model as a "quasi-natural experiment" to verify the impact of the selection of NDZs on urban green innovation. The study found that (1) the NDZ selection policy promotes green innovation in cities and has a significant window-radiating effect. This result still holds after a series of robustness tests. (2) Unlike previous studies that point out that the effect of the policy effect of development zones is related to the level and age of the zones, the results of the heterogeneity test in this paper indicate that the better NDZs and the cities in which they are located, the more conducive to the effect of the national selection policy [7,8]. Specifically, the selection policy is influenced by the characteristics of the city to which it belongs. The larger the city, the more accessible it is, and the more market-oriented it is, the more conducive the national selection policy is to promoting green innovation in the city; the level of government governance also influences the selection policy. The lower level of government governance makes the national selection policy less effective for urban green innovation; the selection policy is also influenced by the level of economic development of the development zone itself. The higher the level of economic development, the more conducive to the effectiveness of the selection policy. (3) The results of the mechanism test show that the selection of NDZs has a positive impact on urban green innovation through the environmental regulation effect, resource allocation effect, and policy amplification effect. To a certain extent, this remedies the existing literature, which only emphasizes the "agglomeration effect" and "selection effect" of development zone policies [5,13]. This paper illustrates the resource allocation effect through capital

and labor, but technology is also an important part of resource allocation, and we believe this is a critical extension study for the future. In summary, NDZ selection policies drive urban green innovation to a certain extent. However, the mechanisms of influence and effects are complex, and this paper makes the following policy recommendations based on these findings.

1. At the regional level, the selection policy for NDZs significantly impacts green innovation in cities and can stimulate green innovation in neighboring cities. This also indicates that upgrading provincial-level development zones to NDZs has significant advantages in operational models, mechanisms, and institutions. Therefore, in the process of policy practice in development zones, China should vigorously exert the exemplary role of NDZs and promote provincial-level development zones to be on par with NDZs, placing equal emphasis on economic and environmental benefits. At the same time, the selection of NDZs should be accelerated. To promote changes in the economic development model, some of the provincial-level development zones with good development momentum and meeting the requirements for upgrading should be upgraded to NDZs, thereby enhancing the overall green innovation level of the city.

2. Given that the effect of national selection policies on urban green innovation is influenced by the characteristics of the development zones themselves and the cities in which they are located, the policies should be promoted in a manner that is appropriate to the local context. Furthermore, it should enhance the flexibility of the policies, effectively assesses the attributes, size, accessibility and marketability of the cities, and comprehensively considers the cities characteristics to develop appropriate policies. It is also vital to improve the level of economic development and government efficiency of the development zones, and to improve the region's "soft environment" so that the national selection policy can better promote green innovation in the city. In addition, the government should also further strengthen the withdrawal mechanism, so that cities whose NDZ policies are not effective can withdraw them scientifically and reasonably to avoid wasting resources.

3. Local governments should shift from GDP-only growth to high-quality development concept of "clear water and green mountains are synonymous to gold and silver mines" (quote from Chinese president Xi). More efforts should be exerted to make full use of the NDZ policy to promote urban green innovation through environmental regulation, resource allocation, and policy amplification effects.

## Acknowledgments

We are very grateful to the journal reviewers for their valuable comments on the manuscript.

## Author Contributions

**Conceptualization:** Xiaolin Yu.

**Data curation:** Xiaolin Yu.

**Formal analysis:** Xiaolin Yu.

**Funding acquisition:** Kai Wan.

**Investigation:** Xiaolin Yu.

**Methodology:** Xiaolin Yu.

**Project administration:** Xiaolin Yu.

**Writing – original draft:** Kai Wan.

**Writing – review & editing:** Xiaolin Yu.

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
