## [Decision Letter · Decision Letter 0]

11 Feb 2022

PONE-D-21-36329How does the National Development Zone policy affect urban green innovation? — E vidence from ChinaPLOS ONE

Dear Dr. WAN,

Thank you for submitting your manuscript to PLOS ONE. After careful consideration, we feel that it has merit but does not fully meet PLOS ONE’s publication criteria as it currently stands. Therefore, we invite you to submit a revised version of the manuscript that addresses the points raised during the review process.

Please see detailed comments from the academic editor and reviewers attached below.

We look forward to receiving your revised manuscript.

Kind regards,

Yanyan Gao

Academic Editor

PLOS ONE

Journal Requirements:

Additional Editor Comments (if provided):

Dear Dr. Kai Wan,

Both reviewers has finished the review upon your submission. While they admit some merits of the paper, some major concerns are also proposed. I agree with them about these concerns. Specifically, I agree with both reviewers that the heterogeneity effect analysis and the mechanism analysis need to be carefully justified, based on the existing literature, and they should stick close to the main effect you investigated. Meanwhile, the literature review should be revised to make it more concise and relevent, and the language should be careful re-organized and polished. Therefore, I would invite you to carefully and substantially revise you paper based on these valuable comments, point by point, which will be sent for the second round review.

Yours sincerely,

Yanyan Gao

Academic Editor

Reviewers' comments:

Reviewer's Responses to Questions

**Comments to the Author**

1. Is the manuscript technically sound, and do the data support the conclusions?

Reviewer #1: Yes

Reviewer #2: Yes

2. Has the statistical analysis been performed appropriately and rigorously? 

Reviewer #1: No

Reviewer #2: Yes

3. Have the authors made all data underlying the findings in their manuscript fully available?

Reviewer #1: Yes

Reviewer #2: Yes

4. Is the manuscript presented in an intelligible fashion and written in standard English?

Reviewer #1: No

Reviewer #2: Yes

5. Review Comments to the Author

Reviewer #1: This paper has evaluated the NDZ policy’s role in affecting urban innovation, using panel data of China’s prefecture-level cities from 2000 to 2018. The article has a clear logical structure and a rigorous argumentative process, but there are still improvements to be made.

1. The authors need to condense the abstract, especially for heterogeneous results, for example “the effect is statistically insignificant in traditional industrial cities and those with underdeveloped transportation”, this is not necessary. The authors have described it from another perspective.

2. The authors need to comb the literatures on NDZ or green innovation, which lacks a large body of classic literature in recent years. Some scholars studied the impact of National Sustainable Development Pilot Zones on economic development. What types of NDZs are included? Is National high-tech industrial development zones an NDZ?

3. What does H1a mean? What is the authors trying to convey? What data did the authors use to verify H1a?

4. The authors need to polish the language of the article to minimize the redundant words, such as so far, in general, ....

5. The authors said “The NDZ policy promotes urban green innovation through resource allocation effect”. The authors analyze the resource allocation effect from the perspective of high-end talents, why not analyze it from the perspective of capital and technology? Capital and technology are also two very important working channels. However, in the mechanism test section, the authors test these two working channels.

6. The heterogeneity analysis of the article is insufficient, the heterogeneity analysis of transportation infrastructure is rather far-fetched, and the author's explanation is difficult to convince the reader. Do the types of development zones and the governance capacity of local governments also need to be considered?

7. In the first paragraph of the conclusion, where is "(2)"? The policy implications are not very relevant and the author needs to further summarize and refine.

Reviewer #2: The topic of this paper is very attractive. After reading it carefully, I found that this paper has some reference value for evaluating National Development Zone policy, but there are some points that need to be optimized. My specific comments are as follows:

1. The authors discuss some of the logical problems in the main text. The following are some examples, but not comprehensive:

"As public good, green innovation suffers the common flaws that can cause market failure, and common practice taken by different countries is policy intervention."

At the start of this paper,the authors discussed the relationship between greem innovation and market failure. As an acedamic paper.,this discussion is too simple and vague.I hope the authors can take more explanation.

2.This paper proposed three paths to support mechanism analysis.But unfortunately, the authors didn’t give an adequate explanation.Why choose these three paths?As mentioned in the article, there are many other paths (e.g. industrial agglomeration, structural optimization...).In my opinion, resource allocation is the best way.Environmental regulation is too routine, and the amplification effect of policy lacks correlation.But I know it's not easy to find good mechanic paths.Therefore, the author can keep these three paths and explain them more.

3.Compared to other productive activities,innovation need more time.In my opinion,this paper should consider adding lag effects to the model.

4.This paper need do more heterogeneity tests to prove your opinions. For example, this paper can further explore the heterogeneity of government governance capacity, heterogeneity of city scale，heterogeneity of marketization level and so on.

5. The result discussions are relatively weak; in particular, the authors need to make some necessary comparison work with previous relevant research, to show the new findings in this paper.

6. PLOS authors have the option to publish the peer review history of their article (what does this mean?). If published, this will include your full peer review and any attached files.

Reviewer #1: No

Reviewer #2: No

---

## [Author Response · Author response to Decision Letter 0]

21 Mar 2022

Reply to the Comments of our paper entitled “How does the National Development Zone policy affect urban green innovation?—Evidence from China” submitted to PLOS ONE(Manuscript ID: PONE-D-21-36329)

We would like to thank the reviewers for their valuable comments mainly on the sections of literature review, theoretical mechanisms, heterogeneity analysis and conclusion suggestions. Based on the suggestions, the author refined the abstract section and the language of the article. In response, substantial changes were made to remedy the deficiencies, as follows.

Reviewer #1

1.The authors need to condense the abstract, especially for heterogeneous results. For example, the sentence “the effect is statistically insignificant in traditional industrial cities and those with underdeveloped transportation” is not necessary to point out since the authors have described it from another perspective.

Re:As you pointed out, our abstract section is not sufficiently concise, and we fully agree with you. In response to your suggestion, we simplified and proofread the text as well as the vocabulary throughout the thesis. There are two important changes as follows.

(1)The abstract is simplified. The first sentence clearly states the key role of the present article in studying the selection policies of NDZs, then followed by an overview of the main content and conclusions of this thesis.

(2)In the introduction, we clearly bring forward the research questions of the article, followed by a detailed description of the entry point and starting point of the article to study the selection policy of NDZs. See the section in red on page 1 of the article for details.

2.The authors need to sort through the literature on NDZ or green innovation, which has lacked a large body of classic literature in recent years. Some scholars has studied the impact of National Sustainable Development Pilot Zones on economic development. What types of NDZ are included? Are National high-tech industrial development zones considered as NDZ?

Re:Your comments are very important and it is true that the original article did not go more deeply into the literature related to the development zone policy and green innovation. We apologize that the original thesis did not give a full description of the types of development zones in China and the ones that are the focus of this paper. In response we made the following changes.

(1)Based on your expert advice, we reorganized the literature on development zone policy and then highlighted the contributions of the present study on this basis. We found that a large number of earlier studies evaluated the policy effects of development zones with different economic functions in China, including economic and technological development zones, high-tech industrial development zones, etc. By comparing regions with and without development zones, it was found that development zones were more advantageous as a location-based industrial policy in terms of economic development and attracting foreign investment (Demurger et al. 2002; Cheng and Kwan, 2000; Akinci and Crittle, 2008). Most of the subsequent studies use the establishment of development zones as a policy shock and explore the impact of development zone policies by using a double difference approach to develop discussions at the firm and regional levels while most of the studies affirm the positive impact of development zone policies (Schminke and Van, 2013; Huang et al., 2017; Luo et al., 2015; Wang, 2013. Alder et al., 2016). However, a small number of scholars have questioned this (Zheng et al., 2016; Mendoza and Valerio, 2016). This paper argues that the reason for this disagreement lies in the fact that established research have overlooked the role of policy selection. Observing the practice process of development zones, it can be found that the upgrading policy of provincial development zones launched in 2009 is its most obvious feature, i.e., regions first pass the provincial government’s approval to become provincial development zones, and then submit applications to the central government, which are then selected as NDZs. Compared to provincial-level development zones, NDZs are larger in scale, serve more national-level development strategies, and have obvious advantages in terms of tax incentives and government subsidies. This suggests that the establishment of a development zone and the selection of a development zone are two completely different processes, with the establishment of a development zone being a process from scratch and the selection of a NDZ being a process of selection among the established provincial-level development zones. For this reason, unlike previous studies, this paper focuses on the impact of national-level development zone selection policies. See the revised section of the article in red for details.

(2)Your comments are very important and we recognised that the literature on green innovation has not been sufficiently organized and for this reason we have added to it. We found established studies that consider green innovation activities as a combination of environmental protection and economic development as an important method for achieving sustainable development (Schiederig et al., 2015; Huang and Li, 2017; Sun et al., 2019). Scholars mostly focused on environmental regulation (Porter and Van-Der-Linde, 1995; Jaffe et al., 1995; Song et al., 2018), government systems (Bai et al., 2018), foreign investment (Lin and Chen, 2018), and the level of economic development (Feng et al. 2017) and then proposed to improve the level of green innovation, as the government system being considered as an important factor driving green innovation in cities. However, few studies have been conducted to uncover the impact and mechanisms of action on urban green innovation from the perspective of NDA selection policies as an important institutional arrangement in China. Our study provides a useful addition to the literature that explores the factors influencing green innovation. See the red section on page 2 of the article for details. 

(3)As you point out, the original article does not indeed state very clearly the types of development zones in China. We appreciate your question as to whether the National Hi-Tech Industrial Development Zone is a NDZ. To this end, we have distilled and clearly explained in the theoretical section. We also provide here a brief description of the development of China’s NDZs, which have played an important role in China’s reform and opening-up process as an important regional industrial policy. In 1984, China first approved the establishment of national-level economic and technological development zones in 14 coastal cities, including Tianjin, Shanghai and Dalian. As of 2018, China has established 552 NDZs and 1,991 provincial-level development zones. While development zones are blossoming everywhere in China, their types are gradually enriched, including economic and technological development zones, high-tech industrial development zones, export processing zones, comprehensive bonded zones, border cooperation zones, etc. Among them, state-level economic development zones and state-level high-tech zones hold an overwhelming advantage in terms of economic contribution to the province. In 2019, China’s 169 state-level high-tech zones and 218 state-level economic development zones achieved a total regional GDP of RMB 22.9 trillion, which is equal to the combined GDP of Guangdong Province and Jiangsu Province in the same year. Therefore they are the key development zones observed in this paper. Looking at the development process of China’s development zones, we can see that in 1984, national-level economic and technological development zones were established in coastal cities open to the outside world with the aim of attracting foreign direct investment, developing private economies, promoting exports, and fostering knowledge-intensive and technology-intensive industries. The State Council has since approved the establishment of national-level economic development zones in other coastal cities and inland open cities one after another. Between 1993 and 2002, every provincial capital city in China set up a national-level economic development zone, while governments at all levels also set up development zones of different levels, resulting in a blossoming of development zones. This has also led to frequent problems such as encirclement of arable land, illegal land concessions and excessive preferences. Therefore, the state started to clean up and consolidate the development zones between 2003 and 2008, and after the completion of the consolidation work, the “China Development Zone Audit Bulletin Catalogue (2006 Edition)” was released. The year 2009 was a key point in the development of development zone policy. The State Council relaunched the upgrading of provincial development zones in 2009, so the regions set off a frenzy to participate in the selection of NDZs. In 2018, the “Catalogue of China’s Development Zones Audit Bulletin (2018 Edition)” was published. NDZs are selected in accordance with five selection criteria, including economic development, scientific and technological innovation, intensive conservation, ecological protection and social responsibility, and provincial-level development zones that meet the criteria are upgraded to national-level development zones, of which scientific and technological innovation and ecological protection are important criteria for the selection of NDZs. See specifically the red section of the article on theoretical foundations and assumptions. In summary, based on your suggestions, we have made several revisions to the article, supplemented by an analysis of some classic literature and the historical process of China’s development zones. We hope to clearly show the research progress related to this paper and the focus objects of this paper.

3. What does H1a mean? What is the authors trying to convey? What data did the authors use to verify H1a?

Re:We are sorry that the original text did not explain the meaning of H1a clearly, this part is indeed ambiguous, and after careful consideration we have considered this part redundant and we decided to remove it and rewrite the theoretical part of the article.

4. The authors need to polish the language of the article to minimize the redundant words, such as so far, in general, ....

Re: We strongly agree with your views. And based on your suggestions, we have streamlined the article, paying special attention to the concise and logical presentation of the text, reducing redundant words, and re-touching the language. We hope that we can highlight the core conclusions while ensuring the integrity of the article’s content, so as to highlight the key points and improve readability. The above changes are scattered, so they are not all marked in red.

5. The authors said “The NDZ policy promotes urban green innovation through resource allocation effect”. The authors analyze the resource allocation effect from the perspective of high-end talents, why not analyze it from the perspective of capital and technology? Capital and technology are also two very important working channels. However, in the mechanism test section, the authors only test these two working channels.

Re:Your suggestions were crucial to our research. And inspired by them, the resource allocation effect generated by the selection policy of NDZs is re-discussed in our theoretical section. Thus we focused on two aspects: capital and labour.

(1)We fully agree with your view that the upgrading of development zones to “national” status will help cities to move up the FDI and industrial capital ladder. And when the capital element is combined with other high-end elements, it can help cities to offer new options for their economic development paths and reduce their dependence on traditional natural resource extraction. The concentration of capital elements can, on the one hand, reduce the transaction costs within the urban green innovation system and, on the other hand, solve the problem of insufficient input of capital elements for green innovation.It helps enterprises to reduce the cost of using capital and improve the efficiency of R&D, thus promoting a higher level of green innovation. Ultimately it is able to bring new drivers to the city, achieve a more advanced and balance development path thus promoting a higher level of green innovation in the city.

(2)We believe that labour mobility is essentially a Pareto-improvement process and will be based on price, supply and demand, competition and other mechanisms to “vote with your feet”. The successful selection of a NDZ will act as a magnetic pole for the labour force and help to spread knowledge and experience among green innovation agents. On the one hand, the agglomeration of labour elements contributes to the phenomenon of “learning by doing” and “learning by learning” in the development zone, which promotes green innovation in the city through knowledge diffusion and spillover (Malberg and Maskell, 1997). On the other hand, with the dual incentive of the NDZ selection and talent introduction policy, it can attract more high-end workforce to the city, which can help enhance the economic growth momentum, thus promoting the city’s transformation of economic growth model and improving the city’s green innovation level. The selection of NDZs can therefore promote green innovation in cities by facilitating the flow of capital and labour factors and optimising the allocation of resources. For details, please see the article revision section.

(3)In line with your suggestion, we also examine the resource allocation effects of the selection policy for NDZs in the empirical section for both capital and labour. We find that the interaction coefficients of both capital and labour allocation with the selection of NDZs are significantly positive, indicating that the selection of NDZs can optimize the allocation of local resources by attracting capital and labour factors, reshaping the core competitiveness of cities and achieving high-quality development. At the same time, we agree with you that technology is also an important part of the resource allocation effect, and we also think that research in this area could be an important task in itself in the future and could produce many valuable results. However, in terms of empirical evidence, it may be necessary to use data from development zones matched to the city level, which will have some technical issues and require a lot more detailed work. Given the amount of work still required to process and construct these data, the empirical estimates based on them also require more careful consideration of identification. We therefore beg your understanding and hope to illustrate this through both capital and labour. However, in the concluding section of the article we add this point, emphasising that this is a very important extension study for the future.

6. The heterogeneity analysis of the article is insufficient. The heterogeneity analysis of transportation infrastructure is rather far-fetched, and the author's explanation is difficult to convince the reader. Do the types of development zones and the governance capacity of local governments also need to be considered?

Re:We are very grateful for your suggestions! Your suggestion would indeed be a fuller and more scientifically sound analysis of the heterogeneity section. We have made the following changes in response to your specific suggestions.

(1)We fully agree with you that the different types of development zones may affect the implementation of development zone policy effects, and Alder et al. (2016) also find that development zone policies have better economic effects when implemented in higher tiered development zones. As a result, we believe that better developed development zones may have a unique advantage, which may be further amplified by the NDZ selection policy, and thus have a differential impact on the level of green innovation in different cities. For this reason, this paper uses the average annual industrial value added density to measure the development level of development zones and divides the sample into high development water development zones and low development development zones. We found that development zones with high levels of economic development are more effective in raising the level of green innovation in the city when they are upgraded to NDZs, while the green innovation effect of the national selection policy was not effective in development zones with lower levels of economic development. Please see Table 8 and the analysis section of the article for details.

(2)Inspired by your suggestion, we have considered the impact of heterogeneity in the level of government governance and used the 2016 China Local Government Management Effectiveness Ranking (jointly published by Management Watch and the Institute of Government Management of Beijing Normal University) to classify provincial government efficiency according to intermediate rankings. A government efficiency ranking in the top 50% is defined as a high level of government governance, otherwise it is a low level of government governance. We find that when the government efficiency ranking is in the top 50%, the NDZ selection policy has a significant effect on urban green innovation. The effectiveness of the NDZs selection policy is difficult to achieve when the government is ranked in the bottom 50% in terms of efficiency. Please see Table 7 and the analysis section for details.

(3)Your comments make us realize that this paper has indeed not been comprehensive enough in terms of heterogeneity analysis. In addition, we would also like to discuss with you that in terms of the transport infrastructure of the development zones, we consider that the improvement of the transport infrastructure is conducive to the spatial mobility of various innovation factors, which is important for reducing costs. And it can also attract more enterprises and capital to the city and optimise the investment environment. Banerjee (2012) and Donaldson (2018) also point out that the opening of highways and the construction of railways (or high-speed trains) can help to reduce trade costs and improve economic efficiency. However, the level of transport infrastructure development in development zones varies depending on the level of urban development. We believe that transport infrastructure, as the hard environment of a development zone, can to some extent influence the relationship between NDZ selection policies and green innovation. For this reason, we prefer to retain the analysis on transport infrastructure heterogeneity in the main text, which we have also modified accordingly in order to enhance completeness and logic.

7. In the first paragraph of the conclusion, where is “(2)”? The policy implications are not very relevant and the author needs to further summarize and refine.

Re:Your suggestion has made us realise that there are existing problems in the conclusions and recommendations section of the article. We summarized and refined this conclusion and recommendation section, then rewrote it. We have made the following main changes.

(1)Inspired by your suggestion, we present the main research work of the article in the conclusion section. We focus on the underlying regression results, the analysis of heterogeneity and the analysis of mechanisms, and compare it with existing studies to highlight the contribution of this paper. The main findings: NDZ selection policies can promote urban green innovation and have a significant window-radiating effect; heterogeneity test results show that the implementation of development zone selection policies in non-old industrial cities, large and medium-sized cities, cities with convenient transportation, and high market-oriented cities can better promote urban green innovation. At the same time, the higher the level of government governance and the better the economic development of the development zone, the more it helps to achieve the effects of the selection policy; the results of the mechanism test show that the selection of NDZs has a positive impact on urban green innovation through environmental regulation effect, resource allocation effect and policy amplification effect.

(2)Based on your suggestions, we have closely aligned the policy recommendations section with the conclusions and made recommendations in the following three main areas: Firstly, the government will vigorously exert the demonstration role of NDZs and accelerate the selection of NDZs. Secondly, the policy should be promoted in a manner that is appropriate to local conditions and enhances the flexibility of the policy. In addition, the government should also further strengthen the exit mechanism, and for cities where the effect of the policy on NDZs is not obvious, achieve a scientific and reasonable exit of the policy to avoid wasting resources. Finally, the pace of reform of the assessment system of local governments should be accelerated, and the traditional concept of “GDP” should be further transformed. At the same time, the market-based system should be accelerated and the government’s ability to intervene in resource allocation should be weakened through decentralisation. We should make full use of the advantages of NDZs to promote the environmental regulation effect, resource allocation effect and policy amplification effect, thereby enhancing the level of green innovation in cities.

Reviewer #2

1.The authors discuss some of the logical problems in the main text. The following are some examples, but not comprehensive: “As public good, green innovation suffers the common flaws that can cause market failure, and common practice taken by different countries is policy intervention.” At the start of this paper, the authors discussed the relationship between green innovation and market failure. However, this discussion is too simple and vague as an academic paper. I hope the authors can take more explanation.

Re:Thank you very much for your valuable comments! In the light of your comments, we have revised the article to fill in the gaps. We have reorganised the logic of the article to avoid overly simplistic discussions. After careful consideration we have removed the reference to market failure due to green innovation from the beginning of the article and have included a discussion and explanation in the first paragraph of the theoretical section. We have also added a discussion of existing research on green innovation in the introduction, which you can find in red in the introduction. The positive externalities, high risks, high inputs and long cycles of green innovation have resulted in a lower rate of return for corporate green innovation than the social rate of return (Arrow, 1962), which to a certain extent reduces the incentive for corporate green innovation and triggers a market failure dilemma. Industrial policy, as an important way for governments to compensate for market failures (Kashyap et al, 2000), can correct the positive externalities of insufficient green innovation and bring the level of green innovation to the social optimum. In addition, inspired by your suggestions, we have further examined and sorted out other parts of the article to rationalise the points raised and to try to make the language more concise and rigorous. We have also reworked the article to make it more readable.

2.This paper proposed three paths to support mechanism analysis. But unfortunately, the authors didn’t give an adequate explanation. Why choose these three paths? As mentioned in the article, there are many other paths (e.g. industrial agglomeration, structural optimization...). In my opinion, resource allocation is the best way. Environmental regulation is too routine, and the amplification effect of policy lacks correlation. But I know it’s not easy to find good mechanic paths. Therefore, the author can keep these three paths and explain them more.

Re:We fully agree with your views and, in response to your comments, we have made a number of changes to the article in order to highlight the theoretical mechanisms highlighted in the article and to add an analysis of the reasons for choosing the three mechanisms for analysis, as well as to provide further explanations of the policy amplification effect.

(1)We follow your suggestion and explain in depth the reasons for choosing three mechanisms for analysis. We first discussed the characteristics of the selection policy for NDZs and concluded that, compared to the establishment of NDZs, the selection policy has the following characteristics. Firstly, the administrative level has been raised. NDZs, in addition, are mostly on a par with other municipal districts and have higher financial rights, whereas provincial-level development zones are merely a street or town under the jurisdiction of a district. Secondly, policy incentives have increased. While provincial-level development zones in China enjoy land tax exemptions, those selected as NDZs are able to enjoy national-level incentives such as VAT exemptions and access to more policy pilot opportunities. Finally, the brand advantage is enhanced. Compared to provincial development zones, NDZs are more scarce, and having a “national” sign is more conducive to attracting capital and talent to the city.

(2)As you have pointed out, the choice of mechanism should be fully explained. To this end, we further explain the selection of the three mechanisms in this paper in terms of the characteristics of the national selection policy. Firstly, under China’s fiscal decentralization and performance appraisal system, NDZs are precisely the economic function areas for local governments to attract investment, and local governments have a great incentive to upgrade provincial-level development zones. In order to meet the ecological protection and innovation conditions selected for NDZs, the most direct means for local governments is to strengthen environmental regulations, and this “competition for the top” can have a catalytic effect on green innovation water in cities (Bu and Wagner 2016; Holzinger and Sommerer 2011). Second, after provincial-level development zones are selected as national-level developments, the business environment of the zones is further optimised. At the same time, better infrastructure and preferential policies will attract high-quality enterprises to enter, accelerating the flow of innovation factors and technology diffusion and improving the allocation efficiency of resources (Lu et al, 2019; Ossa, 2015). Finally, when provincial development zones are successfully upgraded to NDZs, the strength and scope of preferential policies will be further expanded, usually with generous tax incentives and government subsidies for enterprises in the zones, as well as priority treatment in terms of project approval, bank loans, labour organisation and industrial land. A number of policy supports can provide resources to support high-investment, high-risk green innovation activities, which can contribute to the transformation of urban growth and enhance the level of green innovation. In summary, this paper argues that NDZs selection policies can promote urban green innovation through environmental regulation effects, resource allocation effects and policy amplification effects.

(3)Your suggestion is quite valuable. The original article does under-explain the policy amplification effect in the theoretical section and lacks relevant statement. To this end, we have again combed through the policy amplification effect and provided some empirical evidence from China and rewriting this part of the theory. We categorised the impact of national selection policies on urban green innovation through policy amplification effects into two areas. For one thing, as national and provincial development zones are at different levels, their approval bodies and management units may also differ, resulting in different government resources being available to enterprises in the region. Successful selection of national-level development zones will gradually align their policy advantages and development strategies with the national level, giving them more independent authority, simpler approval procedures, and easier access to national support for major projects, thus allowing them to play a “leading role” in regional development (Alder et al., 2016). At the same time, following the major overhaul of development zones, the central government has required the construction of state-level development zones to gradually shift to a new model of industrial optimisation, quality and efficiency, innovation-driven and stable development. Under this development strategy, enterprises will also focus more on organic growth and green innovation, which in turn will lead to green innovation in the city as a whole. Secondly, successful national-level development zones will enjoy greater tax incentives and government subsidies, which are important factors in promoting green innovation in cities. Taking tax incentives as an example, provincial-level development zones in the same province will adopt differentiated tax policies depending on the size of the enterprises, for example, the income tax rate for production-based foreign investment enterprises in Cangzhou Economic Development Zone in Hebei Province is reduced by 24%, while there are no corresponding tax incentives in Xingtai Economic Development Zone. However, once selected as a NDZ, the income tax rate for foreign enterprises in the zone will be reduced by a uniform 15% under the central government, and certain tax incentives will be applied to specific industries and new entrants, further amplifying the policy benefits. This will not only effectively reduce R&D costs for companies (Ohashi, 2005; Clausen, 2009), but also efficiently spread the risk of green R&D activities (Amezcua et al., 2013), thus promoting green innovation in the city as a whole.

3.Compared to other productive activities, innovation needs more time. In my opinion, this paper should consider to add the lag effects to the model.

Re:As you point out, innovation takes more time than other productive activities and should be added to the text more rigorously as a lag effect. For this reason, we further supplement the robustness test by treating the NDZ selection policy with a one-period lag to reflect the lagged effect on urban green innovation. It can be found that the NDZ selection policy has a positive impact on urban green innovation with a significant spatial spillover effect, which also indicates that the lagged effect holds and the conclusions of this paper remain robust. Please refer to Table 10 and the analysis section for details.

4.This paper needs do more heterogeneity tests to prove your opinions. For example, this paper can further explore the heterogeneity of government governance capacity, heterogeneity of city scale, heterogeneity of marketisation level and so on.

Re: Thank you very much for your valuable comments on our heterogeneity analysis section, with which we fully agree. Inspired by your suggestions, We have added substantially about the heterogeneity analysis, which we hope will make the heterogeneity analysis of the paper more comprehensive The details are as follows.

(1)We grouped the sample according to the level of government governance in accordance with your comments, and classified the efficiency of provincial governments according to intermediate rankings by using the 2016 China Local Government Management Effectiveness Ranking jointly published by Management Watch and the Institute of Government Management of Beijing Normal University. A government efficiency ranking in the top 50% is defined as a high level of government governance, otherwise a low level of government governance. The results of the test are reported in Table 7 and it can be found that when the government efficiency ranking is in the top 50%, the NDZ selection policy has a significant contribution to green innovation in the city.

(2)We fully agree with you that we classify cities as large cities, medium cities and small cities according to the Notice on the Criteria for the Classification of City Size issued by the State Council of China. It has been found that in medium and large cities, NDZ selection policies can have a positive impact on urban green innovation, while the effects of selection policies are difficult to achieve in smaller cities. As you point out, our empirical tests have found that city size does affect the relationship between NDZ selection policies and urban green innovation.

(3)Based on your suggestions, we further analysed the heterogeneous impact arising from the level of marketisation and we defined regions with a marketisation index above the median as high marketisation cities and otherwise as low marketisation cities, based on the 2018 China Marketisation Index Report by Regions co-sponsored by the China National Economic Research Institute and Social Science Literature Publishing House. The green innovation effect of the NDZ selection policy was found to be significant in high-market cities and insignificant in low-market cities. The results of this empirical study are also consistent with the reality in China: the top-ranked national-level economic development zones and national-level high-tech zones are mostly concentrated in the eastern coast and other regions with a high level of marketisation. In cities with a low level of marketisation, where economic operation and resource management are mostly subject to the intervention of local governments, the role of NDZ policies will be limited, making it difficult to effectively play a role in promoting green innovation in cities.

5.The result discussions are relatively weak. In particular, the authors need to make some necessary comparison work with previous relevant research for showing the new findings in this paper.

Re: Thank you very much for this suggestion. The article should provide a full discussion of the findings and compare them with existing research, then demonstrating the new findings of the paper. To this end, we have rewritten the conclusions and recommendations section with the following main changes.

(1)We follow your suggestions and discuss the findings of the study. Firstly, our first conclusion clarifies the importance of the NDZ selection policy, which we believe can promote green innovation in cities and has a significant window-radiating effect, a result that continues to hold after a series of robustness tests.

(2)As you have pointed out, a comparison with existing studies is needed in the conclusion section to highlight our contribution. In our second conclusion, we point out that unlike previous studies that focusing on the correlation between the effect of development zone policy effects and the level and age of the development zone (Alder et al., 2016; Wang, 2013), The results of the heterogeneity test in this paper show that the better the development of the development zone and the city in which it is located, the more favorable the effect of the national selection policy. Specifically, the selection policy is influenced by the characteristics of the city to which it belongs. Cities that are not old industrial cities and are larger, more accessible, and more market-oriented are more conducive to national selection policies that promote urban green innovation. The selection policy is also influenced by the level of government governance, with lower levels of government governance making the national selection policy less effective in promoting urban green innovation. Meanwhile, the selection policy is also affected by the level of economic development of the development zone itself, with the higher the level of economic development, the more conducive the selection policy is to achieving its effects, which is a useful addition to existing research.

(3)Our third conclusion focuses on the results of the mechanism test, namely that the selection of NDZs has a positive impact on urban green innovation through environmental regulation effects, resource allocation effects and policy amplification effects. To some extent, it remedies the existing literature’s single mechanism test that only emphasize the “agglomeration effect” and “selection effect” of development zone policies (Schminke and Van, 2013C; Huang et al., 2017). In addition, we have made corresponding recommendations based on the conclusions, which we hope it has the possibility to enhance the logic and integrity of the article.

***Please see red area in our revised version of paper. The literature involved is marked at the end of the text.***

---

## [Decision Letter · Decision Letter 1]

25 Apr 2022

How does the National Development Zone policy affect urban green innovation? — Evidence from China

PONE-D-21-36329R1

Dear Dr. Yu,

We’re pleased to inform you that your manuscript has been judged scientifically suitable for publication and will be formally accepted for publication once it meets all outstanding technical requirements.

Kind regards,

Yanyan Gao

Academic Editor

PLOS ONE

Additional Editor Comments (optional):

Reviewers' comments:

Reviewer's Responses to Questions

**Comments to the Author**

1. If the authors have adequately addressed your comments raised in a previous round of review and you feel that this manuscript is now acceptable for publication, you may indicate that here to bypass the “Comments to the Author” section, enter your conflict of interest statement in the “Confidential to Editor” section, and submit your "Accept" recommendation.

Reviewer #1: (No Response)

Reviewer #2: All comments have been addressed

2. Is the manuscript technically sound, and do the data support the conclusions?

Reviewer #1: (No Response)

Reviewer #2: Yes

3. Has the statistical analysis been performed appropriately and rigorously? 

Reviewer #1: (No Response)

Reviewer #2: Yes

4. Have the authors made all data underlying the findings in their manuscript fully available?

Reviewer #1: (No Response)

Reviewer #2: Yes

5. Is the manuscript presented in an intelligible fashion and written in standard English?

Reviewer #1: (No Response)

Reviewer #2: Yes

6. Review Comments to the Author

Reviewer #1: (No Response)

Reviewer #2: In response to the proposed revisions, the author responded item by item and made serious revisions, effectively solving some of the potential problems in the paper. After this round of revision, I think the paper has reached the publication level of this journal.

7. PLOS authors have the option to publish the peer review history of their article (what does this mean?). If published, this will include your full peer review and any attached files.

Reviewer #1: No

Reviewer #2: No

---

## [Editor Report · Acceptance letter]

28 Jun 2022

PONE-D-21-36329R1 

How does the selection of National Development Zones affect urban green innovation?-Evidence from China 

Dear Dr. Yu:

I'm pleased to inform you that your manuscript has been deemed suitable for publication in PLOS ONE. Congratulations! Your manuscript is now with our production department. 

Kind regards, 

on behalf of

Dr. Yanyan Gao 

Academic Editor

PLOS ONE